# *Ac/Ds*-like Transposon Elements Inserted in *ZmABCG2a* Cause Male Sterility in Maize

**DOI:** 10.3390/ijms26020701

**Published:** 2025-01-15

**Authors:** Le Wang, Saeed Arshad, Taotao Li, Mengli Wei, Hong Ren, Wei Wang, Haiyan Jia, Zhengqiang Ma, Yuanxin Yan

**Affiliations:** 1State Key Laboratory of Crop Genetics and Germplasm Enhancement, Nanjing Agricultural University, Nanjing 210095, China; 2020201074@stu.njau.edu.cn (L.W.); maliksaeedawan@hotmail.com (S.A.); 2020101012@stu.njau.edu.cn (T.L.); wmengli@foxmail.com (M.W.); hyjia@njau.edu.cn (H.J.); ww1980666@126.com (Z.M.); 2Guizhou Institute of Upland Food Crops, Guizhou Academy of Agricultural Sciences, Guiyang 550001, China; rhong666@163.com (H.R.); wwmaize@126.com (W.W.); 3Jiangsu Collaborative Innovation Center for Modern Crop Production, Nanjing Agricultural University, Nanjing 210095, China

**Keywords:** maize (*Zea mays*), seed production, *ZmABCG2a*, mutant *ms*-N125*, mutant *ms*-P884*

## Abstract

Using male sterile (MS) lines instead of normal inbred maternal lines in hybrid seed production can increase the yield and quality with lower production costs. Therefore, developing a new MS germplasm is essential for maize hybrid seed production in the future. Here, we reported a male sterility gene *ms*-N125*, cloned from a newly found MS mutant *ms*-N125*. This mutant has an underdeveloped tassel that showed impaired glumes and shriveled anthers without pollen grains. The MS locus of *ms*-N125* was mapped precisely to a 112-kb-interval on the chromosome 5. This interval contains only three candidate genes, *Zm958*, *Zm959*, and *Zm960*. Sequencing results showed that only candidate *Zm960* harbored a 548-bp transposable element (TE) in its 9th exon, and the two other candidate genes were found to have no genetic variations between the mutant and wild type (WT). Thus, *Zm960* is the only candidate gene for male sterility of the mutant *ms*-N125*. In addition, we screened another recessive MS mutant, *ms*-P884*, which exhibited similar male sterility phenotypes to *ms*-N125*. Sequencing *Zm960* in *ms*-P884* showed a 600-bp TE located in its 2nd exon. *Zm960* encodes an ATP-binding cassette in the G subfamily of ABC (ABCG) transporters, *ZmABCG2a*, with both mutants which harbored an *Ac/Ds*-like transposon in each. To verify the function of *ZmABCG2a* for male sterility further, we found an ethyl methanesulfonate (EMS) mutant, *zmabcg2a**, which displayed male sterility and tassel phenotypes highly similar to *ms*-N125* and *ms*-P884*, confirming that *ZmABCG2a* must be the gene for male sterility in maize. In addition, the results of lipid metabolome analysis of *ms*-N125* young tassels showed that the total lipid content of the mutant was significantly lower than that of the WT, with 15 subclasses of lipids, including PE (phosphatidylethanolamine), PC (phosphatidylcholine), DG (digalactosyldiacylglycerols), and MGDG (monogalactosyldiacylglycerol) which were significantly down-regulated in the *ms*-N125* mutant versus its wild type. In summary, we identified alternate mutations of the *ZmABCG2a* gene, which may be a potential germplasm for hybrid seed production in maize.

## 1. Introduction

As an important cereal, maize (*Zea mays* L.) has become the most extensively cultivated crop worldwide, providing vital support for global food security [1,2]. The heterosis of maize significantly enhances yield, seed quality and resistance to diseases [3]. Using heterosis in maize is one of the most successful examples of this crop production strategy [4]. Detasseling is a critical step in maize hybrid seed production. However, manual detasseling causes incomplete removal of tassels and high costs of seed production [5]. Utilizing male-sterile (MS) lines for hybrid seed production can significantly reduce production costs and improve the purity percentage of the hybrid seed product [6]. Therefore, developing maize MS lines will substantially benefit the maize seed industry [6,7].

Male sterility (MS) is a natural phenomenon in plants. Maize can be classified into two main categories [8]: cytoplasmic male sterility (CMS) and nuclear male sterility, also known as gene male sterility (GMS). CMS in maize, including CMS-T, CMS-S, and CMS-C, is strongly associated with a number of issues such as a limited number of restorer lines, unstable male sterility influenced by environmental conditions, and susceptibility to southern corn leaf blight caused by *Bipolaris maydis*. Therefore, CMS application in maize hybrid seed production has been minimized over the past forty decades [9]. In this scenario, GMS has become the only practical strategy for hybrid seed production in maize.

During the early stages of crop heterosis research, utilization of GMS has been widely unrecognized due to no maintenance line being available for GMS. In 2006, scientists at DuPont Pioneer designed a new technology called “Seed Production Technology” (SPT), which significantly increased the potential of GMS in maize hybrid seed production [10,11]. The crucial point of SPT is to create a GMS maintainer, which can provide pollen grains to homozygous MS plants to obtain vast volumes of MS line seeds for hybrid seed production. Maintainers can be self-propagated by self-pollinating. Notably, the SPT system did not include transgenic elements in hybrid seeds [10,11]. Later on, in 2018, a new version of SPT, a multi-control sterility system (MCS), was developed [12]. The MCS system significantly enhanced the efficiency of the maize MS line propagation for commercial hybrid seed production. In addition, an updated version of MCS, the dominant genetic male sterility (DGMS) system, was established and which was argued to be suitable for various crops such as maize and rice [13].

Maize anther development is a multi-layered biological process. Anther development is a complex biological process, including many biological events such as signal transduction, cell division, apoptosis, substance transportation, and metabolism. During anther development, approximately 24,000 to 32,000 genes are involved in this event, and defects in some of these genes often lead to GMS [14,15].

Up to date, more than 30 GMS genes have been cloned in maize [16], and they have been classified into five functional categories: (1) Lipid metabolism and transportation. For instance, maize genes *ZmMS2* and *ZmMS44* encode lipid transfer proteins in transporting lipids between cells [17,18]. Other genes, such as *ZmMS5/ipe2* [19], *ZmMS10/apv1* [20], *ZmMS20/ipe1* [21], *ZmMS26* [22], *ZmMS30* [23], *ZmMS33* [24], and *ZmMS25/ms-6021* [25], encode GDSL class lipases, cytochrome P450 mono-oxygenases, and glycerol-3-phosphate acyltransferases. These genes directly participate in fatty acid metabolism within anther tissues and in the formation of large lipid molecules like cutin, wax, and sporopollenin [26]. (2) Transcription factor (TF) genes involved in anther and other floral organ development. A set of TFs regulates the development of the tapetum layer in maize anthers. RNA-seq data predict that over 1100 TFs participate in maize tapetum development, including *ZmMS23*, *ZmMS32*, *ZmbHLH122*, and *ZmbHLH51*. These TFs establish a proper proliferation pattern in maize tapetal cells [27]. (3) Carbohydrate metabolism genes. Bioinformatics and RNA-seq analyses predicted 112 maize carbohydrate metabolism genes associated with *ms40*, with most showing peak expression during late anther development [28]. (4) Small RNA genes. Small RNAs, including *ms28* [29], *ocl4* [30], and *dcl5* [31,32], play a role in anther development. (5) Other genes. For example, *MSCA1* encodes a glutathione peroxidase [33,34].

In our breeding materials we noticed that two MS mutants, *ms*-N125* and *ms*-P884*, showed typical MS phenotypes. In this study, we isolated an MS gene in these two mutants by position-based cloning. We showed that *ms*-N125* encodes an ABCG-type transport protein, ZmABCG2a, which plays a crucial role in anther wall and cuticle formation during anther development. *ms*-N125* and *ms*-P884* are alleles of the same gene, ZmABCG2a, which means they are located in the same locus on the chromosome but have different mutations. Our work found two *ZmABCG2a/MS13* mutants which might be alternative maternal lines for maize hybrid seed production via SPT technology.

## 2. Results

### 2.1. ms*-N125 Presented Typical Male Sterility (MS) Phenotypes

Compared with wild-type (WT) plants, *ms*-N125* showed similar healthy somatic growth and female fertility. However, the development of the male spike was abnormal as it withered prematurely. The spikelet structure was normal, but the anthers could not protrude and were atrophied and empty. In addition, there were no viable pollen grains, leading to complete male infertility (Figure 1A).

We used paraffin sectioning to analyze the cytological structures of anthers at various stages in fertile and sterile plants within the BC_2_F_2_ population. The objective was to identify the abnormalities in the anthers of mutant *ms*-N125* and determine the time of abnormal anther development.

Maize anther development is classified into 14 stages (S1 to S14) based on cytological structures [6]. Under an optical microscope, we observed that the *ms*-N125* microspore mother cells and anther wall layers remained identical to the WT until stage S6 (Figure 1B). At the beginning of stage S7, *ms*-N125* anthers showed abnormalities such as persistent middle layers, darker and thicker tapetal cells, and delayed tapetal breakdown compared to the WT anthers. At stages S8a and S8b, these abnormalities resulted in fused dyads and tetrads, whereas stage S9 marked the beginning of abnormal microspore development. The middle layer persisted until stage S12, resulting in complete male infertility in maize pollen grains.

### 2.2. Genetic Mapping and Gene-Cloning of ms*-N125

F_1_ and F_2_ populations resulting from the cross between *ms*-N125* and B73 were used for genetic analysis. Through backcrossing with B73, all F_1_ progeny exhibited a normal phenotype. A chi-square test in the F_2_ generation revealed that the segregation ratio adhered to a 3:1 (fertile/sterile) ratio (χ^2^ = 0.9164 < 3.841), indicating that a single recessive gene causes the male sterility mutation in *ms*-N125*.

In this study, the Euclidean distance (ED) method was used for positioning analysis, and the positioning region of *ms*-N125* was determined according to the association threshold. A total of 398,402 original SNP and InDel loci were obtained in this study, and 200,984 high-quality SNP and InDel loci were obtained after filtering. A BSR-seq association analysis was carried out on these high-quality SNP loci to obtain the loci closely associated with the target traits. The depths of each base in the two pools of the wild type and the mutant were counted, and thus, the ED values of each locus were calculated. The fifth power of the original ED value (ED^5^) was taken as the association value, and then the LOESS method was used to fit all the ED values. The association threshold for the analysis was set as the median plus three times the standard deviation (median + 3SD) of the fitted values of all loci. The threshold in this study was 0.9989764. Based on the association threshold, it could be determined that one associated region located on chromosome 5 of maize was obtained (Appendix A). A confidence interval was determined on chromosome 5 of maize, specifically between chr5:19,527,705 thymine/cytosine (T/C) and single nucleotide polymorphism (SNP)-40,899,235 guanine/adenine (G/A), with a physical distance of 21.4 Mb (megabase). This interval contained 491 genes, making it the central region linked to *ms*-N125* (Figure 2A).

To further refine the *ms*-N125* interval, primers were designed within the associated interval, and gene mapping was performed based on genotype–phenotype correlation. The target gene was first mapped with nine indel markers using the F_2_ (712 plants) and F_3_ (627 plants) segregating populations. The gene was initially mapped to an interval of 2.12 Mb between markers indel 2566 and indel 2778. Then, using BC_1_ and BC_2_ populations (1601 plants), fine mapping was carried out between markers indel 2566 and indel 2778 using eight pairs of uniformly spaced primers (Figure 2B, Appendix A).

Between indel 2622 and indel 2633, a minimal distance of 112 kb (kilobases) was determined using four recombinant individual plants. There were three annotated gene models: Zm00001d013958 (candidate 1), Zm00001d013959 (candidate 2), and Zm00001d013960 (candidate 3) within this 112 kb interval based on the annotation of B73 reference genome (B73 RefGen_v4), as shown in Figure 2B. The sequencing of these three genes revealed that Zm00001d013960 contains an hAT-AC transposon insertion in its ninth exon, with an inserted fragment size of 548 bp (Figure 2C, Appendix A). Zm00001d013960 encodes an ATP-binding cassette transporter, also known as *ZmABCG2a* [35].

### 2.3. ms*-P884 Is an Allelic Mutant to ms*-N125

In 2019, another MS mutant was identified in the Hainan experimental station, exhibiting a similar phenotype to the *ms*-N125* (field code: 2020p884) referred to as *ms*-P884* (Figure 3A). Crosses were performed with B73 as the paternal parent with the sterile line *ms*-P884*. The F_1_ progeny exhibited normal fertility, while the F_2_ generation segregated in a 3:1 (fertile/sterile) ratio (χ^2^ = 0.978 < 3.841, Figure 3B), indicating that a single recessive gene causes the male sterility mutation in *ms*-P884*.

After attaining the stable inheritance of the mutant phenotype, we hypothesized that the two mutants, *ms*-P884* and *ms*-N125*, might be allelic variations of the same gene. Allelic tests between *ms*-P884* and *ms*-N125* were performed to test this hypothesis using two combinations: *ms*-P884* (*a*′*a*′) × *Ms*-N125* (*Aa*) and *ms*-N125* (*aa*) × *Ms*-P884* (*Aa*′). The results showed that the progeny of both combinations exhibited a 1:1 (fertile/sterile) ratio (χ^2^ = 0 < 3.841; χ^2^ = 0.008 < 3.841, Figure 3B), suggesting that *ms*-P884* and *ms*-N125* are indeed allelic at the same locus. Consequently, we sequenced the *ZmABCG2a* locus in *ms*-P884* and detected a Trep1169-like transposon insertion in the second exon. The inserted fragment has a size of 600 bp (Figure 3C). Identifying the causative mutation of *ms*-P884* further confirmed that structural variations in *ZmABCG2a* lead to male sterility.

### 2.4. Functional Verification of ms*-N125 via Ethyl Methanesulfonate (EMS) Mutant

We screened an EMS-induced premature stop codon mutant line of *ZmABCG2a,* called *zmabcg2a** (Figure 4A). In addition, we genotyped *zmabcg2a* using restriction enzyme digestion to further support and validate the association between the mutation in *ZmABCG2a* and male sterility (Figure 4B,C). We planted *zmabcg2a** plants and observed their phenotype during the tasseling and pollen-shedding stages. The mutant tassels in the field did not produce any pollen, providing evidence of male sterility of *zmabcg2a** (Figure 4D).

To further authenticate the function of the candidate gene in the *ms*-N125* mutant, we conducted the following allelic tests: *zmabcg2a**(*ms/ms*) × *Ms*-N125* (*MS/ms*) and *zmabcg2a**(*ms/ms*) × *Ms*-P884* (*MS/ms*). The results showed that the progeny of both combinations exhibited a 1:1 (fertile/sterile) ratio (χ^2^ = 0.964 < 3.841; χ^2^ = 0.964 < 3.841, Figure 4E), suggesting that *zmabcg2a**, *ms*-P884*, and *ms*-N125* are indeed allelic at the same locus. These results indicate that the mutation in *ZmABCG2a* leads to male sterility in *ms*-N125*.

### 2.5. The ms*-N125 Tassels Displayed a Significant Decrease in Long-Chain Lipids

Using UHPLC-MS (ultra-performance liquid chromatography-mass spectrometry), we examined the differences in lipid metabolism between fertile tassels and *ms*-N125* at the V13 stage. The tassels of these two materials contained 1088 lipid from 49 lipid subclasses, with glycerolipids, glycerophospholipids, and sphingolipids being the leading lipid derivatives found in the maize tassels (Appendix A). The total lipid molecular contents in the *ms*-N125* tassels were significantly lower than its fertile sibling, decreasing from 4,097,111,596 molecules in fertile siblings to 2,779,723,611 molecules in *ms*-N125*. Thus, the relative lipid contents in *ms*-N125* mutant tassels were only 67.8% of that in fertile siblings (Figure 5A).

While comparing the lipid subclasses between *ms*-N125* and fertile sibling tassels, 42 subclasses showed differences in contents, with 19 of these differences reaching significant levels. Among the 19 lipid subclasses that showed significant differences, 15 were significantly down-regulated in the *ms*-N125* mutant tassels than in its fertile siblings, including BisMeLPA, CmE, Co, D5TG, DG, LPC, LPEt, MG, MGDG, MePC, OAHFA, PC, PE, SiE, and WE. Only four lipid subclasses (AcHexChE, AcHexSiE, AEA, and PG) had higher contents in *ms*-N125* than their fertile siblings. Based on the comprehensive comparison of contents and significance, PE, PC, DG, and MGDG were the four lipid subclasses that showed the most significant differences (Figure 5B, Appendix A).

Furthermore, the chain length and saturation analysis were carried out for the metabolites of the four lipid subclasses: PE, PC, DG, and MGDG. The results showed that these four metabolites’ long-chain lengths (Figure 6A, Appendix A) and saturations (Figure 6B, Appendix A) were significantly reduced. Thus, it indicates that a sequence of changes in the tassel of the lipid carbon chain length and the saturation occurred following the mutation in *ms*-N125*, which could be a factor in male sterility.

### 2.6. The Down and Up-Regulation of Lipids in ms*-N125 from Hierarchical Clustering

Hierarchical clustering utilized the expression levels of significantly distinct lipid molecules. The objective was to observe more clearly the variations in the patterns of lipid molecules expressed in the tassels of *ms*-N125* and fertile sibling plants across several samples (VIP > 1, *p*-value < 0.05). Figure 7 shows that 108 differential lipid molecules were detected between *ms*-N125* and fertile siblings, in which 71 showed down-regulation, and the remaining 37 showed up-regulation (Appendix A).

## 3. Discussion

Transposable elements (TEs) are a significant component of most plant genomes. These mobile repetitive sequences exhibit high diversity in abundance, structure, transposition mechanisms, activity, and insertion specificity. According to sequencing, over 85% of the maize genome comprises TEs. The transposition of TEs can lead to new genetic mutations, resulting in new mutants. These mutants are the most direct and effective genetic materials for identifying new genes, which can significantly enrich germplasm resources. In this study, a spontaneously occurring male sterile mutant named *ms*-N125* was discovered in the field in 2016. The mutation in this variant is caused by insertion of an hAT-*Ac* transposon, measuring 556 bp (base pairs) in length, into the ninth exon of the gene Zm00001d013960. Zm00001d013960 encodes an ATP-binding cassette transporter known as *ZmABCG2a*. Another male-sterile (MS) mutant, *ms*-P884*, was later discovered in 2019 and exhibited a phenotype similar to that of *ms*-N125.* Allelic tests confirmed that *ms*-N125* and *ms*-P884* belong to the same allelic group. An hAT-Tip100-like transposon inserted into the second exon of *ZmABCG2a* causes the mutation in *ms*-P884*. Further validation was provided by the *zmabcg2a** mutant.

The ATP-binding cassette in the G subfamily (ABCG) transporters is one of the largest subfamilies within the ABC (ATP-binding cassette) transporter superfamily, with subfamily members playing critical roles in various processes affecting plant adaptability. These ABCG transport proteins have been involved in essential biological processes, including anther and pollen development, plant and female organ development, biotic and abiotic stress responses, and plant hormone transport and signaling. Functional characterization has been performed for at least 30 ABCG genes in *Arabidopsis* and 11 ABCG genes in rice. However, among the 51 ABCG genes in the maize genome, only the following three have been fully characterized: *ZmGL13* [36], *ZmMs2* [18]/*ZmABCG26* [37], and *ZmMs13* [35]. These three genes have been identified and reported to be involved in maize male fertility or leaf cuticle formation. Specifically, *ZmMs13* was recognized by cloning two mutants, *ms13* and *ms13-6060*, which are typical male sterility genes. A 20-bp insertion and a 3-bp deletion mutation in the *ZmABCG2a* characterize them, respectively. They are located at the same locus as our cloned *ms*-N125*. This gene is closely associated with lipid metabolism in maize tassel and the development of anthers.

Homologous genes of *ZmABCG2a* have been reported both in rice and *Arabidopsis* [38]. In *Arabidopsis*, the most homologous gene to *ZmABCG2a* is *AtABCG11* [38]. Although *AtABCG11* and *ZmABCG2a* are highly homologous, they have significant differences. *ZmABCG2a* is intensively expressed in immature tassel_v13 and meiotic tassel_v18 (Appendix A), which implies its function may be mainly relevant to the tassel development in maize [35]. Furthermore, *zmabcg2a* affects anther cuticle formation, resulting in abnormal microspore development, ultimately leading to pollen-deficient male sterility [39,40].

In contrast, *AtABCG11* gene expression spans the entire reproductive period in *Arabidopsis*. The *atabcg11* mutants exhibit defects in cuticle formation in both vegetative and reproductive tissues, displaying a reduced development rate and dwarf stature, and leading to minimal fertility-related indicators. However, the mutant can still produce viable seeds, classifying them as semi-sterile [39,41]. The difference in the functions of *AtABCG11* and *ZmABCG2a* might be caused by the functional redundancy of homologous genes within maize. While compared with *Arabidopsis*, the rice homolog *OsABCG26* shares a closer resemblance with *ZmABCG2a* [38]. Both genes are specifically expressed in reproductive tissues; however, *zmabcg2a* exhibits delayed and abnormal anther development at stage 8b (Figure 1B). In addition, *osabcg26* postponed the disintegration of the middle layer and irregular microspore development, leading to male sterility [42]. These findings suggest that, as our model indicates, the anther cuticle and pollen exine may share a common lipid synthesis pathway within the tapetum (Figure 8).

ABCG transporters facilitate the formation of the anther cuticle and pollen wall by transporting various cuticular waxes and cutin monomers, sporopollenin precursors, and tryptophan derivatives from the tapetum to the outer surface and interior chambers of the anther [38,42]. Loss-of-function mutations in some ABCG transporters often result in defects in anther cuticle and pollen wall formation, causing male sterility. The *ZmABCG2a* is likely to regulate a pathway consistent with *OsABCG26* and *ATABCG11*. Defective anther cuticle layer formation is the main effect of the functional mutation of *ZmABCG2a,* indicating that *ZmABCG2a* may generate anther cuticle precursors in the tapetum before transporting them out.

Plants undergo extensive metabolic changes during development and in response to environmental signals [43,44]. Understanding the overall reshaping of lipid profiles is crucial for exploring the role of lipids in regulating reproductive organ development in crops. According to lipid content analysis, the relative lipid content of the *ms*-N125* mutants only makes up 68% of the wild type (WT), which reveals the lower total lipid levels in the tassels of the mutants. These lipid molecules may be critical indicators for maize tassel adaptation and survival. Of the top 15 metabolite components that exhibit substantial variations between WT and *ms*-N125* tassels, only two are upregulated, and 13 are downregulated. These differentially expressed metabolites fall into three main categories: glycerolipids (TG, DG, and d5-TG), glycerophospholipids (PE), and sterol lipids (SiE).

Interestingly, these five metabolite types substantially differ in chain length and saturation. Glycerolipids function as storage reservoirs, energy sources, and thermal insulators in eukaryotic cells. However, glycerophospholipids, including PE subtypes, constitute the major lipid structures on eukaryotic cell membranes [45]. On the other hand, sterol lipids are essential components of various tissues and cells, forming lipoproteins and contributing to the composition of cell membranes [46].

Furthermore, in plants, changes in chain length are regulated by relevant lipid-metabolizing enzymes, directly impacting membrane fluidity. Therefore, this affects membrane permeability, substance transport, and the localization function of membrane proteins [47,48,49]. Additionally, chain saturation affects the flexibility of cell membranes, which in turn affects cell division, migration, and signal transduction. These processes are critical for developing disease and stress reactions [50,51,52]. For example, in plants, membrane saturation is regulated through homeoviscous adaptation to cope with temperature changes [53,54]. Hence, these results imply that the *ZmABCG2a* mutation modifies the length and saturation of the chain in maize tassels, which leads to male sterility.

The anther cuticle and pollen exine consist of fatty acids and their derivatives. The lipids play a crucial protective role for the male gametophyte against various environmental stresses. Lipid metabolism is involved in the formation of both these protective barriers. However, abnormalities in lipid metabolism cause genetic male sterility due to unsuccessful microspore development. However, the exact processes that control the movement of lipids from tapetal cells to other tissues are still being investigated.

**Figure 8 ijms-26-00701-f008:**
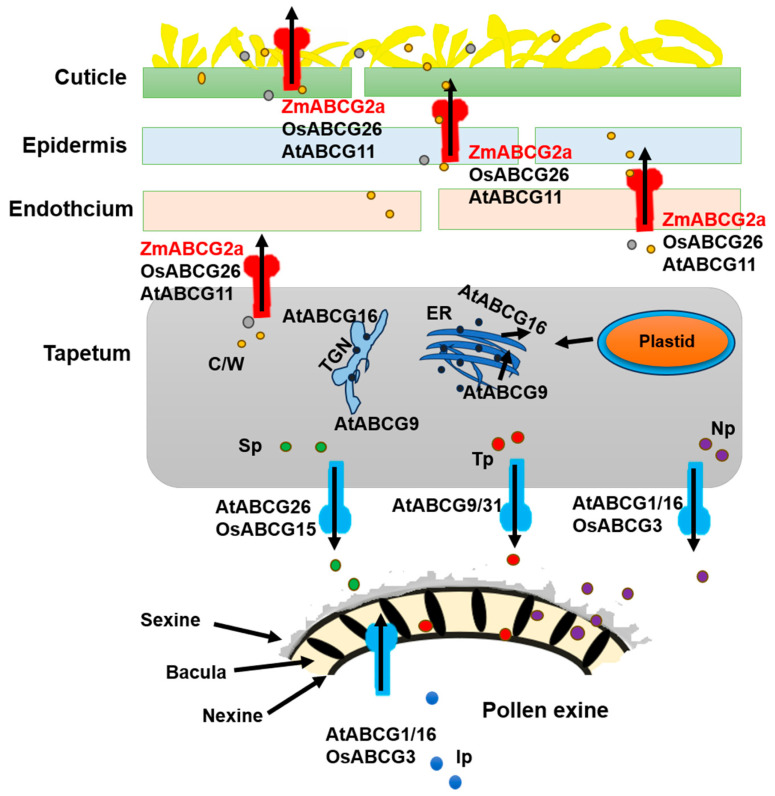
The proposed mechanism of action model ABCG (ATP-binding cassette in the G subfamily) transporters for anther and pollen development (Adapted and edited with permission from Wu et al., 2022 [38]). ABCG transport proteins are involved in various processes, including the transport of cuticular waxes, cutin monomers, pollen protein precursors, and tryptophan components. These ABCG transporters transfer lipids and other precursors produced in the tapetum layer to support the development of the anther cuticle and pollen wall. OsABCG26/ATABCG11/ZmABCG2a is a half-sized membrane-bound transporter among these transporters. It may form homodimers or heterodimers for export. It probably brings petal, endothelial, and epidermal forming cells to the surface during the degeneration of the middle layer in stages S9 and S10, aiding in the development of the anther cuticle. The sporopollenin precursor in the tapetum layer is transported to the anther locule via OsABCG15/ATABCG26. In the context of pollen exine development, *AtABCG9* and *AtABCG31* are involved in the specific transport of sterol glycosides within the tapetum layer, contributing to pollen coat (tryptophan) deposition [55]. In *Arabidopsis*, *AtABCG1* and *AtABCG16*, along with their rice ortholog *OsABCG3,* are essential for connecting protein and protein precursor transport during post-meiotic male gametophyte development and pollen tube growth [56,57,58]. C/W, cutin and wax; Ip, intine precursors; Np, nexine precursors; Sp, sporopollenin precursors; TGN, trans-Golgi network; Tp, tryphine precursors.

## 4. Materials and Methods

### 4.1. Plant Materials and Experimental Locations

The *ms*-N125* line was discovered in autumn 2016 within a tropical lineage of glutinous maize known as N125 in Hainan, China. Male sterility occurred naturally in about 25% of these plants. Likewise, In the autumn of 2020, natural male sterile materials were discovered in the maize materials numbered 2020P884 in the fields of Hainan and named *ms*-P884*. The maize ethyl methanesulfonate (EMS) induced *ms*-N125* premature termination mutant (EMS4-209c46) was obtained from the maize EMSDB (http://maizeems.qlnu.edu.cn/, accessed on 1 November 2024), designated as *zmabcg2a**. The material genetic background of *zmabcg2a** is B73 (the pollen of B73 treated with EMS before pollination). The wild type (WT) used in this research was B73. All maize materials were grown at the Baima Experimental Station of Nanjing Agricultural University and Hainan Experimental Station, Sanya, China, from 2016 to 2023.

### 4.2. Phenotypic Characterization

The morphological characteristics of mature maize and *Arabidopsis* plants were captured using a Canon 200DII digital camera (Tokyo, Japan) and a macro zoom stereomicroscope (MVX10). The anthers were examined using a dissecting microscope (Olympus SZX10, LECO Corporation, St. Joseph, MI, USA). Mature pollen grains from freshly opened anthers were collected on slides, stained with 1% iodine–potassium iodide (I2-KI), and observed under a fluorescence microscope (Olympus) for the viability test.

The samples were collected during the anther development stage for cytological observations. The sampling time was from 8 a.m. to 10 a.m. Well-grown plants were randomly selected, and the spikes were quickly cut open at the parts that felt soft. After removing the tassels that met the requirements, the anthers were stripped from the spikelets with tweezers and anatomical needles and lengths were measured. Immediately, anthers were placed into the corresponding centrifuge tubes according to the plant numbers and anther lengths, respectively. Anthers were fixed in FAA (formaldehyde–acetic acid–alcohol fixative) solution, dehydrated in a gradient ethanol series (50–100%), and embedded in paraffin. Sections were prepared using a microtome (Leica DM2255, Wetzlar, Germany) and stained with Safranin O–fast green staining for observation under a macro zoom stereo microscope [59,60].

### 4.3. Genetic Analysis and Map-Based Cloning of ms*-N125 Locus

For genetic analysis of the *ms*-N125* locus, we hybridized the *ms*-N125* mutant and the standard fertile inbred line B73 to obtain F_1_ hybrid seeds. Subsequently, self-pollination yielded the F_2_ population. Integrating phenotypic traits, we constructed a high-generation backcross segregating population using a “backcross-selfing” model.

From the BC_1_F_2_ segregating population, we randomly selected 50 male sterile and 50 fertile plants. RNA (ribonucleic acid) sample pools containing three biological replicates were constructed. Total RNA was extracted from maize leaf samples at the V3 stage using the TransZol UP Plus RNA Kit (TransGen, Beijing, China). The Beijing Biomarker Biotechnology Co., Ltd. (Beijing, China) was employed to perform BSR-seq (bulk segregant RNA-seq). Genetic analysis, marker design, and map-based cloning of the *ms*-N125* locus were conducted based on published methods and BSR sequencing results [25,61].

### 4.4. Nucleic Acid Isolation

Genomic DNA was extracted from the leaf tissues of an individual plant using the cetyltrimethylammonium bromide (CTAB) method [62].

### 4.5. Lipid Metabolomics Analysis

The tassel spikelets were collected as lipid analysis samples at the anther developing stage (S13). The sampling time was from 9 a.m. to 11 a.m. The spikelets of the tassels were collected and immediately placed into liquid nitrogen [63]. Four biological replicated samples of mutant of *ms*-N125* and WT were sent to the Shanghai Sanshu Biosciences Co., Ltd. for non-targeted lipid metabolomics analysis (https://sanshugroup.com/, accessed on 10 September 2024).

The samples were frozen in liquid nitrogen and homogenized for 3 cycles (each cycle: 5500 rpm for 20 s, repeated 3 times). Then, 400 μL of methyl tert-butyl ether (MTBE) and 80 μL of methanol (MeOH) were added, following vortexing for 30 s and ultrasonication for 10 min and then centrifuged at 3000 rpm for 15 min to separate the phases. The upper MTBE layer was added to a new EP tube for drying (to a fixed volume, such as 200 μL), and evaporated to dry material. After reconstitution (in EP tubes) the dried extract was dissolved in 100 μL of DCM (dichloromethane):MeOH (1:1, *v*/*v*), and 5 μL of each sample was pooled as a QC (quality control) sample and the QC sample measured for every 12 individual samples.

UPLC Conditions: The sample extracts were analyzed using a UPLC-Orbitrap-MS system (UPLC, Vanquish; MS, QE). The analytical conditions were as follows. UPLC: column, Waters HSS T3 (100 × 2.1 mm, 1.8 μm); column temperature, 40 °C; flow rate, 0.3 mL/min; injection volume, 2 μL; solvent system, acetonitrile: water (6:4, 10 mM HCOONH_4_):acetonitrile:isopropanol (1:9, 10 mM HCOONH_4_); Gradient program, 60:40 *v*/*v* at 0–9 min, 0: 100 *v*/*v* at 9.0–22.0 min, 60:40 *v*/*v* at 22.0–26.0 min.

QE (quality evaluation) analysis: HRMS data were recorded on a Q Exactive hybrid Q-Orbitrap mass spectrometer with a heated ESI source (Thermo Fisher Scientific, Waltham, MA, USA) utilizing the Full-Scan-MS2 acquisition methods. The ESI source parameters were set as follows: spray voltage, 3.0 kV/−2.8 kV; sheath gas pressure, 60 arb; aux gas pressure, 10 arb; sweep gas pressure, 0 arb; capillary temperature, 320 °C; and aux gas heater temperature, 350 °C.

Data analysis and interpretation: Data were acquired on the Q-Exactive using Xcalibur 4.1 (Thermo Scientific), and processed using Lipid search. The data of the tassel lipids analysis were obtained, and then analysis of variance (ANOVA) was conducted using the Statistical Package for Social Science (SPSS; SPSS Inc., Chicago, IL, USA) version 22.0. Statistical significance was judged at a threshold of *p* < 0.05.

## 5. Conclusions

The only candidate gene for male sterility identified in the *ms*-N125* mutant was *Zm960*. Additionally, the screened recessive male sterile mutants, *ms*-P884* and *ms*-N125*, exhibited similar phenotypes and characteristics, such as stunted tassels and wrinkled anthers without pollen. The male sterility of *ms*-N125* and *ms*-P884* results from the mutation of the same locus. This candidate gene encodes ZmABCG2a, an ATP-binding cassette of the G subfamily of ABC (ABCG) transporters, which was caused in both mutants by an *Ac/Ds*-like transposon. *ZmABCG2a* plays an important role in the lipid metabolism of maize tassels. This research documented alternative *ZmABCG2a/MS13* mutants that could be applied in the future as maternal lines for hybrid maize seed production.

## Figures and Tables

**Figure 1 ijms-26-00701-f001:**
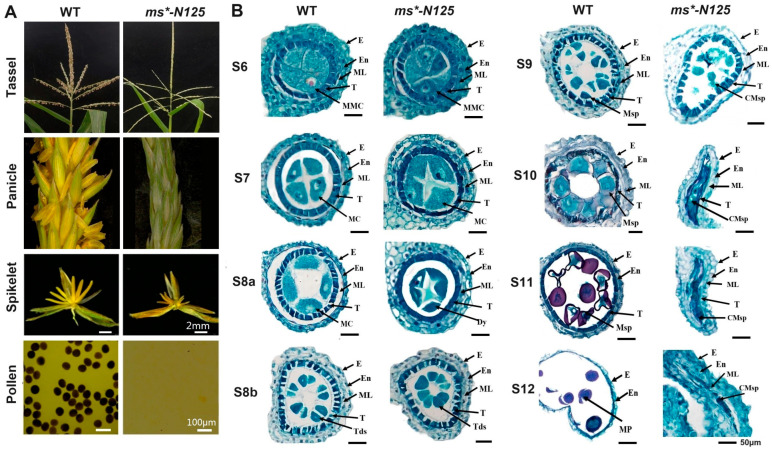
Phenotypic characterization of *ms*-N125* mutant. (**A**) Comparison of the *ms*-N125* male-sterile (MS) phenotypes, those displayed by wild-type (WT) plants. Iodine–potassium iodide (I2-KI) stained pollen grains. (**B**) The comparative anther section of WT and *ms*-N125* at stage 6 (S6), 7 (S7), 8 (S8a/b), 9 (S9), 10 (S10), 11 (S11), 12 (S12) of anther development. The paraffin sections of anthers were stained with safranin/fast green. E, epidermis; En, endothecium; ML, middle layer; MC, meiotic cell; T, tapetum; MMC, microspore mother cell; Dy, dyad; Mp, mature pollen; Tds, tetrads. Msp, microspore; CMsp, collapsed microspore.

**Figure 2 ijms-26-00701-f002:**
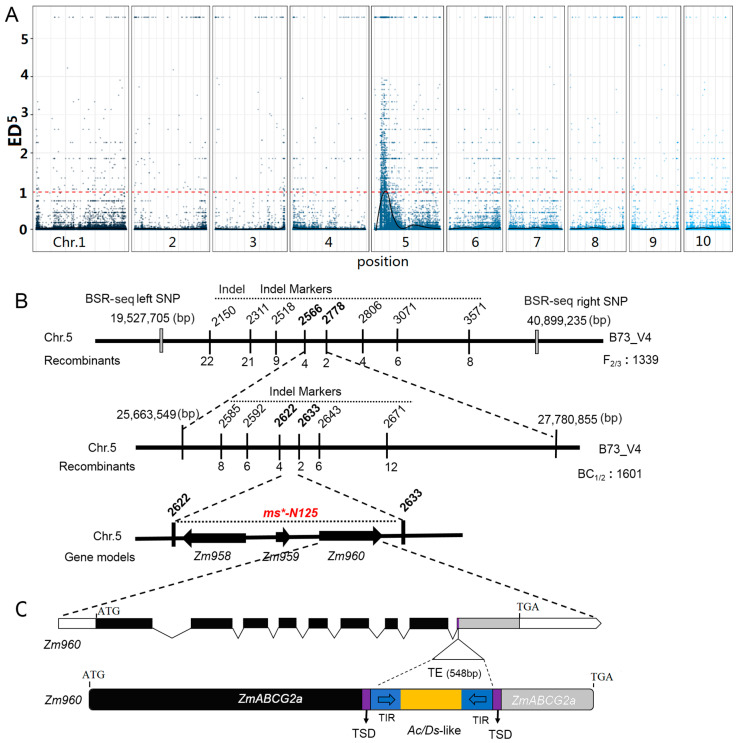
Map-based gene cloning procedure of the male sterile (MS) gene in *ms*-N125*. (**A**) Distribution of ED^5^ (Euclidean distance to the fifth power) values of all single nucleotide polymorphism (SNP) loci across ten chromosomes by bulked segregant RNA-seq (BSR-seq). Blue dots represent ED^5^ values for each SNP locus. The black curve indicates the Loess regression analysis, identifying the region where the mutation of *ms*-N125* exists. The red dashed line represents the significant threshold associated with Loess regression analysis. (**B**) Fine mapping of *ms*-N125* using InDel markers. The final mapping region of the locus *ms*-N125* is a 112 kb (kilobases) interval between InDel 2622 and 2633 on chromosome 5, containing three predicted candidate genes. (**C**) The third candidate gene, *Zm960* (*ZmABCG2a*) in *ms*-N125* mutant, harbors a 600-base pair (bp) transposable element (TE). Black boxes represent the coding sequence of the gene model. The grey box indicated the coding sequence loss by transposon insertion. TSD, target site duplication by TE insertion, is depicted in purple. The TE insertion, an *Ac/Ds*-like transposon was highlighted in yellow color and its terminal inverted repeats (TIR) were shown in blue.

**Figure 3 ijms-26-00701-f003:**
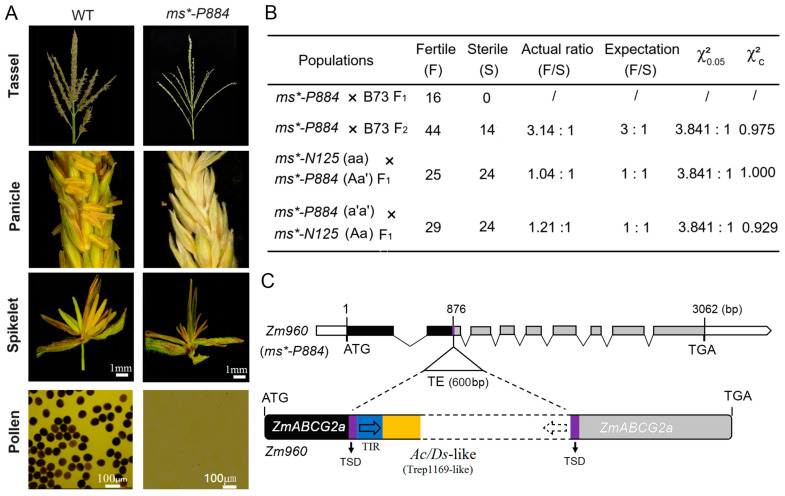
Phenotypic characterization and genetic analysis of male sterile (MS) mutant *ms*-P884*. (**A**) Phenotypic characterization of *ms*-P884* mutant in comparison with wild type (WT). The WT produced viable pollen, stained with I2-KI (iodine–potassium iodide), while the mutant produced no pollen. (**B**) Allelic testing of *ms*-P884* with *ms*-N125*. The letter a represents the recessive mutation of the *ms*-N125* mutant and the letter a’ is the allelic variation of the same gene on the mutant *ms*-P884*. (**C**) *ZmABCG2a* in the mutant *ms*-P884* containing a 600-bp (base pair) transposable element (TE). Black boxes represent the coding sequence of the gene model. The grey box indicates the coding sequence loss by TE insertion. TSD (target site duplication) and TIR (terminal inverted repeats) were shown as that in Figure 2.

**Figure 4 ijms-26-00701-f004:**
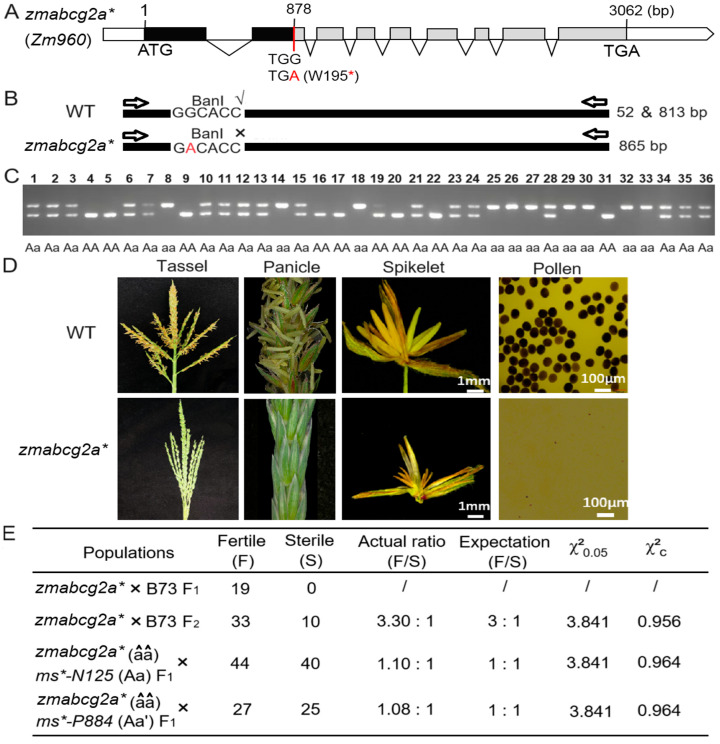
Genetic screening and phenotypic analysis of ethyl methanesulfonate (EMS) induced mutant of *ZmABCG2a* gene. (**A**) Gene structure of *zmabcg2a**, an EMS-induced translation-stop mutant of *ZmABCG2a*. Black boxes represent exons; the red line indicates the EMS-induced nucleotide mutation sites. The red letter A shown the genetic mutation TGG to TGA. Red asterisk means the translation of mRNA was stopped at 195th amino acid due to the point mutation. (**B**) Identification of the mutant site in *zmabcg2a**. (**C**) Genotyping of *zmabcg2a** mutant plants by PCR and restriction digestion with Ban I (G▼GYRCC). The primers are listed in Appendix A. (**D**) Phenotypic characterization of the *zmabcg2a** mutant at stage S13. (**E**) Genetic analysis of the *zmabcg2a** mutant and allelic tests with mutants *ms*-N125* and *ms*-P884*. The letter a, a′ and â represent the mutated gene of *ZmABCG2a* in the mutant *ms*-N125*, *ms*-P884* and *zmabcg2a**, respectively.

**Figure 5 ijms-26-00701-f005:**
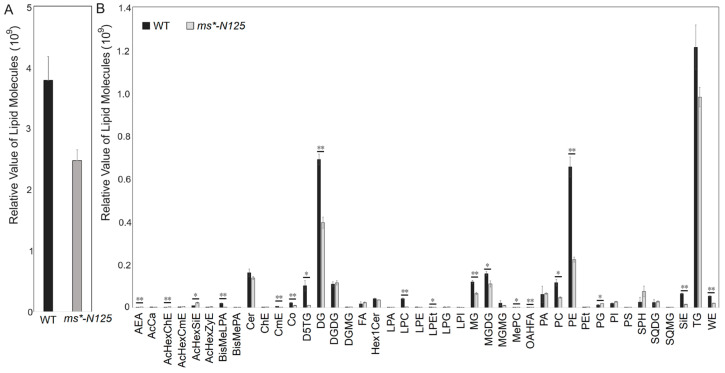
The lipid relative level in the young tassels of mutant *ms*-N125* and its fertile siblings. (**A**) Comparison of the response values for the total lipid molecule content between wild type (WT) and *ms*-N125.* (**B**) Comparison of the differences in lipid subclasses between *ms*-N125* versus WT in maize tassels. The horizontal axis represents each lipid subclass; different colors distinguish different groups. The vertical axis represents the intensity of lipid subclasses, reflecting their relative contents. The asterisks denote significant differences between WT and mutant at *p* < 0.05 (*) or *p* < 0.01 (**) by *t*-test.

**Figure 6 ijms-26-00701-f006:**
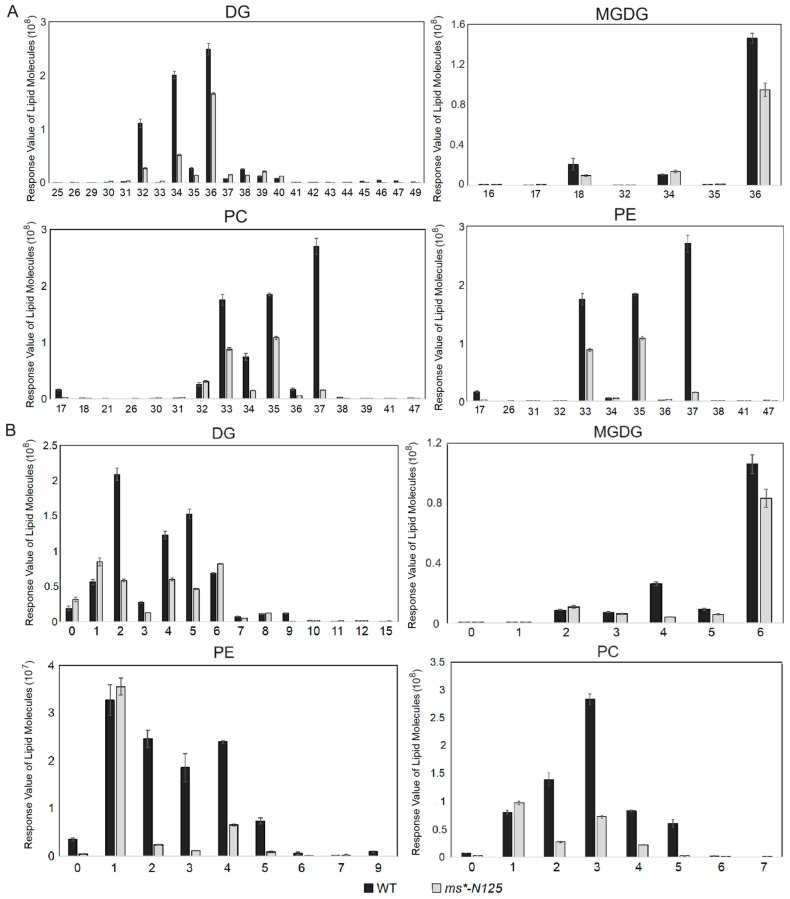
The carbon chain length and saturation analysis of lipid molecules in male sterile *ms*-N125* and fertile siblings. Carbon chain length (**A**) and saturation (**B**) analysis of differentially expressed lipid molecules between tassels of *ms*-N125* and fertile siblings. DG, digalactosyldiacylglycerols; MGDG, monogalactosyldiacylglycerol; PC, phosphatidylcholine; PE, phosphatidylethanolamine.

**Figure 7 ijms-26-00701-f007:**
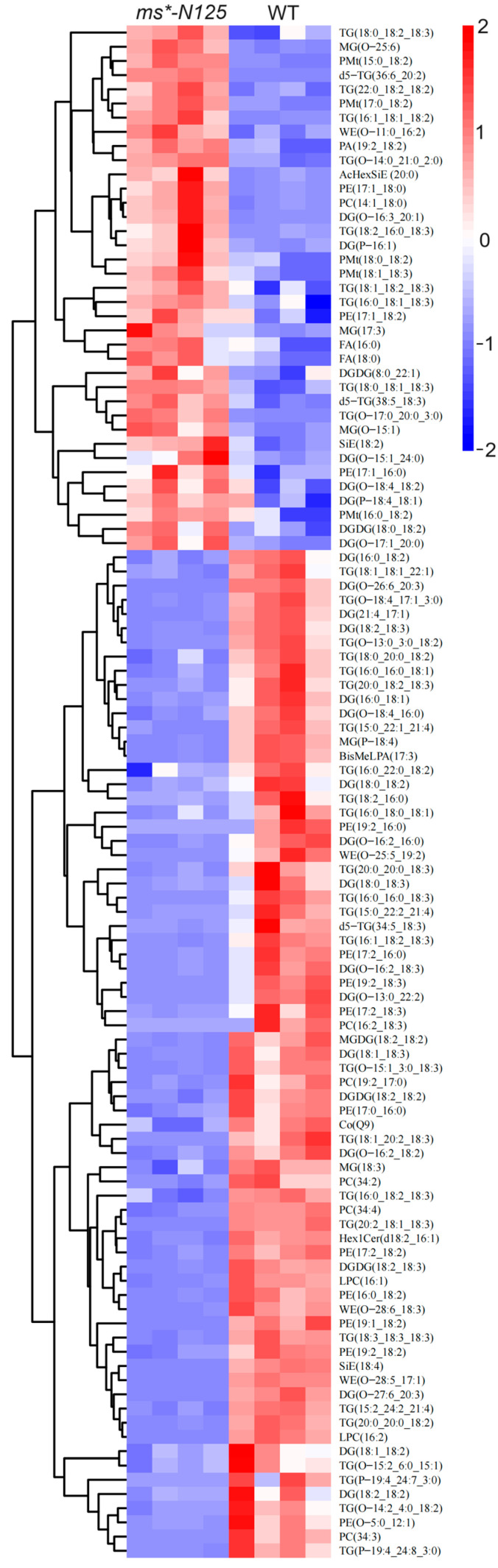
The relative content of differential lipid molecules in mutant and wild-type samples. Each column represents a sample of lipid metabolomics analysis, and each row represents a lipid molecule. The color key indicated the lipid molecules’ relative expression levels in each sample. Red indicates higher expression levels, while blue indicates lower expression levels. The dendrogram on the left shows the clustering of lipid molecules, while closer branches indicate more similar expression levels. The IDs of the lipid molecules are listed on the right, and the names of the samples are listed at the top. The specific information of each lipid molecule in the picture can be found in Appendix A. Differential lipid molecules that satisfied the screening criterion of VIP > 1 and *p*-value < 0.05 were chosen.

## Data Availability

The authors confirm that this study’s data and supporting findings are available within the article and its Appendix A.

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
