# Peer review of "Ac/Ds-like Transposon Elements Inserted in ZmABCG2a Cause Male Sterility in Maize"

_ijms, 2025, doi:10.3390/ijms26020701_

Round 1
Reviewer 1 Report
Comments and Suggestions for Authors
Dear authors,
Very nice study which was interesting to read. I do have a number of suggestions to help improve your manuscript however. Please also use the annotated PDF which I generated as part of my review of your manuscript.
Title - I suggest you rethink your title so that it more accurately describes your work, and increases readership of your article.
Abstract - please see annotated PDF for specifics, but quite a lot of English language issues are present throughout - please amend accordingly. Content of Abstract is good however as it accurately describes your study.
Introduction - English language issues present throughout. Please see annotated PDF and correct accordingly.
- species names should be provide on first mention in brackets are the common name e.g., maize (Zea mays)
- only however consideration is the length of the Introduction - it is currently quite wordy, and therefore word count could be reduced if you see it fit to do so.
Results -main concern is English language issues, please use the annotated PDF to identify each instance and correct accordingly.
Figure 1 is excellent - well done.
Figure 2 is of high quality - well done.
Figure 3 is of high standard - well done
Figure 4 is of a high standard, but maybe consider reducing its size (width and height) as this will present better.
Figure 5 and 6 should be combined together as they report the same results and this will improve the impact of the presented data
Figure 7 is of good standard, but please improve the information in the legend of this Figure.
Figure 8 presents an excellent model - well done. Maybe you could convert this to a graphical Abstract? Then all in-text Figures will only be presenting experimental data? Just a suggestion.
Discussion - very high standard where you have supported your findings in this study by those made by others in either maize of other plant species.
- please address each English language issue identified in the annotated PDF
Conclusion - please highlight all of your excellent findings more clearly in the Conclusion. The current version does not outline the quality of your findings very well. Make it short and sharp and highly focused.

English language correction is required.
Not by an English language editing service, but by the authors themselves
Author Response
Comments 1: Dear authors, Very nice study, which was interesting to read. However, I have several suggestions to help improve your manuscript. Please also use the annotated PDF I generated as part of my review of your manuscript.
Respond 1: We appreciate your help. We revised the manuscript according to your PDF provided and thanks. For your concern points, please see the word file attached.

Reviewer 2 Report
Comments and Suggestions for Authors
This study analyzed the phenotypes and sequences of two maize male-sterile mutants, ms-N125 and ms-P884, revealing similar phenotypic characteristics and the insertion of identical Ac/Ds-like transposon elements in the male-sterility gene Zm960. Combined with EMS mutagenesis analysis, it was demonstrated that Zm960, encoding the protein ZmABCG2a, is the male-sterility gene in maize. ZmABCG2a was shown to complement the Arabidopsis abcg11 mutant, which is deficient in cutin and wax. Additionally, compared to the wild type, the lipid content of ms-N125* was significantly reduced.
Recommendations for improvement:
- Discussion, lines 318: What evidence supports the claim that ZmABCG2a exhibits tissue specificity during the maize growth cycle?
- Lines 246: Have the four upregulated lipid subclasses been specifically analyzed? What is the mechanism behind their upregulation?
- Results, lines 168: The "556 base pairs" mentioned does not align with Fig. 2C, which shows 548 bp. Please verify.
- Results, lines 139: Ensure consistent formatting.
- Fig. 3, panel B: Should an annotation be added to explain the label "a'"?
- Regarding the mechanism linking lipid metabolism to male sterility, is further analysis required? Specifically, how does the mutation in ZmABCG2a impact lipid metabolism and lead to male sterility?
Author Response
Comments 1: This study analyzed the phenotypes and sequences of two maize male-sterile mutants, ms*-N125 and ms*-P884, revealing similar phenotypic characteristics and the insertion of identical Ac/Ds-like transposon elements in the male-sterility gene Zm960. Combined with EMS mutagenesis analysis, it was demonstrated that Zm960, encoding the protein ZmABCG2a, is the male-sterility gene in maize. ZmABCG2a was shown to complement the Arabidopsis abcg11 mutant, which is deficient in cutin and wax. Additionally, the lipid content of ms*-N125 was significantly reduced compared to the wild type.
Respond 1: We really appreciate your comments and suggesions.

Reviewer 3 Report
Comments and Suggestions for Authors
This study basically applied the map-based cloning approach to identify the three MS mutant in maize: ms*-N125, ms*-P884, and zmabcg2a*. The overall conclusions are solid, and work amount are huge. For example, the author combined genetic mapping, lipidomic, and functional validation for robust conclusions. However, I have several major concerns which seems must be fixed before its publication.
First, the author should rewrite the abstract to make sure the logical is reasonable and maintain its fluent.
Second, the author should rewrite several sections from Materials and Methods to make they are complete and sufficient details for the readers. For example, more details on the ED5 metric's significance and calculation would clarify its application in this study. Additionally, the approaches and their details of Fig 1B, Fig 2A, Fig 5, Fig 6 and Fig 7 are either absent or lacking detailed description.
Third, although the author compared some of the discovery difference between the ABCG genes in rice, maize and Arabidopsis, the comparative analysis with other ABCG transporters in maize could deepen insights into functional redundancy or specialization.
Fourth, in discussion, some of statements are Ignoring some evolutionary concepts. For example, L327, “These differences between zmabcg2a and atabcg11 suggest potential functional divergence during evolution”, it obvious that AtABCG11 and ZmABCG2a are not the ortholog pair since the ZmABCG2a shows the male specific function which indicate it is an evolutionary young gene by gene duplication (see a young gene involved in male sterility case: https://academic.oup.com/plcell/article-abstract/28/9/2060/6098364). This means that the ortholog of AtABCG11in maize is other paralogs of ZmABCG2a, but not ZmABCG2a. It is inaccurate for the author to state that “functional divergence”.
1. L18: please provide the detail about the discovery of such material: “a newly found”
2. L23: even within the non-coding region?
3. L25: “another recessive MS mutant, ms*-P884”, please indicate the resource of this mutant since this is important to trace the relationship between the two MS lines.
4. L25: “displayed similar phenotypes”: please provide the details of such statement since this is critical to identify the molecular/phenotypical functions of the candidate genes.
5. L26: “same genetic locus”, show the evidence of such statement.
6. L29: it is logically and practically unacceptable to identify an external EMS mutant to confirm the function of ZmABCG2a which cause a MS phenotype in another MS mutant ms*-N125. Do the two lines have exactly same genetic background? Why not just perform a genetic complementary experiment within the MS mutant ms*-N125? To screen an EMS mutant in a specific gene is huge amount to work and usually not meet the expectation.
Comments on the Quality of English Languagena
Author Response
Comments 1:This study applied the map-based cloning approach to identify the three MS mutants in maize: ms*-N125, ms*-P884, and zmabcg2a*. The overall conclusions are solid, and the work amount is huge. For example, the author combined genetic mapping, lipidomic, and functional validation for robust conclusions. However, I have several major concerns that must be fixed before publication.
Response 1:
Thanks a lot for your comprehensive review. It is revised based on your comments and suggestions. Please find our responses in the file attached.

Round 2
Reviewer 3 Report
Comments and Suggestions for Authors
NA